# One-Year Progression and Risk Factors for the Development of Chronic Kidney Disease in Septic Shock Patients with Acute Kidney Injury: A Single-Centre Retrospective Cohort Study

**DOI:** 10.3390/jcm7120554

**Published:** 2018-12-15

**Authors:** June-sung Kim, Youn-Jung Kim, Seung Mok Ryoo, Chang Hwan Sohn, Dong Woo Seo, Shin Ahn, Kyoung Soo Lim, Won Young Kim

**Affiliations:** Department of Emergency Medicine, Asan Medical Center, University of Ulsan College of Medicine, Seoul 05505, Korea; jsmeet09@gmail.com (J.-s.K.); yjkim.em@gmail.com (Y.-J.K.); chrisryoo@naver.com (S.M.R.); schwan97@gmail.com (C.H.S.); leiseo@gmail.com (D.W.S.); ans1023@gmail.com (S.A.); kslim@amc.seoul.kr (K.S.L.)

**Keywords:** septic shock, acute kidney injury, acute kidney disease, chronic kidney disease, follow-up

## Abstract

(1) Background: Sepsis-associated acute kidney injury (AKI) can lead to permanent kidney damage, although the long-term prognosis in patients with septic shock remains unclear. This study aimed to identify risk factors for the development of chronic kidney disease (CKD) in septic shock patients with AKI. (2) Methods: A single-site, retrospective cohort study was conducted using a registry of adult septic shock patients. Data from patients who had developed AKI between January 2011 and April 2017 were extracted, and 1-year follow-up data were analysed to identify patients who developed CKD. (3) Results: Among 2208 patients with septic shock, 839 (38%) had AKI on admission (stage 1: 163 (19%), stage 2: 339 (40%), stage 3: 337 (40%)). After one year, kidney function had recovered in 27% of patients, and 6% had progressed to CKD. In patients with stage 1 AKI, 10% developed CKD, and mortality was 13% at one year; in patients with stage 2 and 3 AKI, the CKD rate was 6%, and the mortality rate was 42% and 47%, respectively. Old age, female, diabetes, low haemoglobin levels and a high creatinine level at discharge were seen to be risk factors for the development of CKD. (4) Conclusions: AKI severity correlated with mortality, but it did not correlate with the development of CKD, and patients progressed to CKD, even when initial AKI stage was not severe. Physicians should focus on the recovery of renal function, and ensure the careful follow-up of patients with risk factors for the development of CKD.

## 1. Introduction

Sepsis is one of the most common causes of mortality in critically ill patients worldwide [1,2,3]. Septic shock, the most severe form of sepsis, can lead to multi-system organ failure and is a major risk factor for the development of acute kidney injury (AKI), accounting for more than 50% of cases [4,5]. Although septic AKI has been considered a temporary syndrome [5,6], a growing body of evidence suggests that AKI is likely to lead to continuous or permanent kidney damage, and it can progress to end-stage kidney disease [7,8]. In patients with sepsis and septic shock, the presence of AKI has been shown to be a poor prognostic factor that is associated with higher rates of mortality and short-term adverse consequences, including the prolonged duration of mechanical ventilation, increased intensive care unit stay and death [9,10]. However, few studies have included long-term follow-up periods [11,12], and data describing the relationship between initial severity and the development of chronic kidney disease (CKD) in patients with sepsis-induced AKI are limited [13].

To address this issue, we evaluated data from the Asan Medical Center Emergency Department Septic Shock Registry, to determine the development of CKD in septic shock patients with AKI, and to identify risk factors associated with the development of this condition.

## 2. Materials and Methods

### 2.1. Setting and Study Population

This single-center, retrospective, observational, registry-based study was conducted at the Asan Medical Center Emergency Department in South Korea, using data obtained from patients diagnosed between January 2011 and April 2017. The Asan Medical Center is an academic tertiary referral center with 2700 beds; approximately 100,000 patients visit the emergency department annually. The study protocol was approved by the institutional research ethics committee (Study No. 2016-0548), and the requirement for informed consent was waived, due to the retrospective nature of the study.

Adult patients (≥18 years of age) with septic shock were enrolled from the Asan Medical Center Septic Shock Registry. Septic shock was defined as the presence of refractory hypotension (mean arterial pressure ≤70 mmHg) requiring treatment with vasopressors, or a blood lactate concentration ≥4 mmol/L despite sufficient fluid loading [14]. We excluded individuals who were younger than 18 years and pregnant individuals. Moreover, in addition to evaluate the effect of newly developed AKI to CKD, we also excluded patients who had previously been diagnosed with CKD and end-stage renal disease (ESRD) requiring renal replacement therapy (RRT).

### 2.2. Data Collection and Definition

Data regarding patient age, sex, previous medical history, laboratory results and infection sites based on clinical and radiological examination were obtained from the registry. Previous CKD or ESRD patients who had outpatient or inpatient diagnosis of pre-existing CKD and ESRD, had code related with hemodialysis and who had a prior diagnosis of AKI or a baseline-estimated glomerular filtration rate (eGFR) lower than 60 mL/min/1.73 m^2^ were identified via electronic medical records. AKI on initial admission was defined in accordance with the Kidney Disease: Improving Global Outcomes (KDIGO) guidelines [15], i.e., an increase in serum creatinine (Cr) of 0.3 mg/dL within 48 h, or an increase in creatinine to 1.5 times the lowest known creatinine level during the preceding one week to one year. If baseline Cr levels were not available, an estimated baseline was calculated using the simplified modification of diet in renal disease (MDRD) formula, assuming that a given patient without known renal disease had a normal glomerular filtration rate (GFR) of approximately 75–100 mL/min/1.73 m^2^. Serum Cr levels were assessed in all patients at least once each day during the hospital stay, and baseline, initial, peaks within 48 h and discharge values, were recorded. Maximum KDIGO refers to the worst KDIGO stage observed over the 48 h period following admission. Baseline Cr levels were measured preceding one week to one year before admission, and the initial level was measured at admission.

To evaluate the CKD status after one year, serum Cr and eGFR levels were obtained after discharge from the electronic medical records of all patients, 12 ± 3 months from initial admission. Moreover, in order to reduce missing diagnoses, we collected data of inpatient or outpatient diagnoses of CKD or ESRD and codes related to hemodialysis, via medical records. All eGFRs were calculated by MDRD (GFR = 175 × serum Cr^−1.154^ × age^−0.203^ × 1.212 (if patient is black) × 0.742 (if female)). When multiple records were present, the highest serum Cr and the lowest GFR values were recorded. CKD risk (based on GFR values, mL/min/1.73 m^2^) was then classified according to the KDIGO guidelines: G1 ≥90, G2 = 60–89, G3a = 45–59, G3b = 30–44, G4 = 15–29 and G5 <15 [16]. Patients classified as G3a, G3b, G4 and G5 were included in the analysis; those with G1 and G2 disease were excluded, as they were considered to be at a low risk of developing ESRD. The date of the patient’s death was extracted from the National Health Insurance Service in South Korea. The primary study outcome was the development of CKD according to the initial and the maximum KDIGO AKI stage. Secondary outcomes included all-cause mortality and RRT dependence within the 1-year follow-up period.

### 2.3. Statistical Analyses

Statistical analyses were performed using SPSS Statistics for Windows, version 23 (SPSS Inc., Chicago, IL, USA). Continuous variables were reported as the median and interquartile range. Categorical variables were analysed using the chi-square test or Fisher’s exact test. The normality of distribution was examined using the Kolmogorov–Smirnov test. The Mann–Whitney *U* test was used for the comparison of CKD and non-CKD groups after one year of follow-up. Variables with an entry-level significance of *p* < 0.2 in the univariate analysis were included in a stepwise multivariate analysis, because an entry-level of less than 0.2 was more informative than that of 0.1. Possible interactions and collinearities were also tested. To adjust for confounding variables, and to assess possible effect modification, separate multiple logistic regression analyses were performed. The results were reported as odds ratios (OR) and 95% confidence intervals (CI). A *p*-value <0.05 was considered to be statistically significant.

## 3. Results

### 3.1. Patient Characteristics

Between 1 January 2011 and 31 April 2017, 2208 adult patients were enrolled in the Asan Medical Center Emergency Medicine Septic Shock Registry (Figure 1). Of these, 255 who had a pre-existing diagnosis of CKD or ESRD, and 1114 patients with septic shock without AKI, were excluded. The remaining 839 patients (38%) with AKI were categorised according to their KDIGO classification on the day of admission. Among these 839 patients, 163 (19%) had stage 1, 339 (40%) stage 2 and 337 (40%) stage 3. According to the maximum KDIGO criteria, 117 (14%) had stage 1, 337 (40%) stage 2 and 385 (46%) stage 3. Within the first 48 h, maximum serum creatinine was recorded and 35 patients were reassigned from stage 1 to stage 2; 48 patients were additionally included in the stage 3 group. Among them, 151 patients applied continuous RRT during admission, and six patients needed intermittent hemodialysis after stopping continuous RRT.

The demographic, clinical and laboratory characteristics of patients who developed/did not develop CKD after one year (CKD and non-CKD groups) are summarised in Table 1. Overall, patients were predominantly male (63.3%), with a median age of 64 years. Hypertension and diabetes were more common in the CKD group than the non-CKD group (47.4% vs. 32.8%, *p* = 0.045; 52.6% vs. 23.6%, *p* < 0.001, respectively). No significant differences were seen in other underlying diseases between the two groups. Pulmonary (21.6%) and hepatobiliary (30.9%) infections were most common in both groups, and no statistically significant differences were seen in the locations of the infection sites. Regarding laboratory values, baseline, initial, peak and discharge creatinine, blood urea nitrogen and hemoglobin levels tended to be higher in the CKD group than in the non-CKD group.

### 3.2. KDIGO Stages and Outcomes

Clinical outcomes in the CKD and non-CKD groups are shown in Figure 1. Among the 117 stage 1 patients, 15 (13%) died, 42 (36%) were lost to follow-up, 48 (41%) recovered full kidney function and 12 (20%) developed CKD (KDIGO CKD stage G3a, *n* = 8 patients; G3b, *n* = 3; G5, *n* = 1). Of the 337 stage 2 patients, 140 (42%) died, 86 (26%) were lost to follow-up within one year, 91 (27%) recovered full kidney function and 21 (6%) developed CKD (stage G3a, *n* = 7; G3b, *n* = 11; G4, *n* = 2; G5, *n* = 1). Of the 385 stage 3 patients, 181 (47%) died, 89 (23%) were lost to follow-up, 90 (23%) recovered full kidney function and 24 (6%) developed CKD (stage G3a, *n* = 8; G3b, *n* = 8; G4, *n* = 5; G5, *n* = 3).

The adjusted ORs of the initial and maximum KDIGO AKI stage for CKD development and all-cause mortality within 1 year are shown in Table 2. Notably, there were no significant differences in the occurrence of CKD by KDIGO classification between the initial and maximum criteria. The OR for CKD development according to the AKI stages increased proportionally, but it was not statistically significant. Meanwhile, all-cause mortality proportionally increased according to KDIGO classification by the initial and maximum Cr levels.

### 3.3. Risk Factors for the Development of CKD

A multivariate logistic regression of factors associated with the occurrence of CKD development within one year is shown in Table 3. Older age (adjusted OR: 1.070, 95% CI: 1.033–1.108, *p* < 0.001), diabetes (adjusted OR: 2.620, 95% CI: 1.352–5.078, *p* = 0.004), low hemoglobin levels (adjusted OR: 0.840, 95% CI: 0.744–0.949, *p* = 0.005) and higher discharge creatinine levels (adjusted OR: 2.686, 95% CI: 1.499–4.812, *p* < 0.001) were associated with the development of CKD.

## 4. Discussion

In this study, we evaluated the development of CKD in patients with septic shock-associated AKI. In patients who survived and for whom 1-year follow-up data were available, 80% (229/286) recovered full renal function within 1 year and 20% (57/286) had progressed to CKD; 2% (5/286) were dependent on RRT. Long-term all-cause mortality was 40% (336/839), and while the severity of the KDIGO AKI stage was correlated with mortality, it did not correlate with the development of CKD.

In the current study, the incidence of AKI in patients with septic shock was 38% on admission, which is consistent with previous studies reporting an incidence of approximately 35% [17]. AKI can accelerate the progression of CKD [7,18,19], but little is known about the association between AKI severity and the development of CKD. Ishani et al. showed that CKD developed in 6.6%–10.5% of elderly patients with AKI [8]. In addition, they found that elderly individuals, particularly those with previously diagnosed CKD, were at significantly greater risk for end-stage renal disease, suggesting that episodes of AKI may accelerate the progression of renal disease [8].

Few studies have assessed the relationship between AKI severity and CKD progression, and this association therefore remains the subject of some debate [20,21,22]. Chawla et al. hypothesised that the severity of AKI, according to the Risk, Injury, Failure, Loss and End-stage kidney disease (RIFLE) criteria, correlates with the progression of CKD [23]. In addition, a recent meta-analysis reported that the risk of CKD increased proportionally with mild, moderate and severe AKI (adjusted hazard ratio: 2.0, 3.3 and 28.2, respectively) [18]. However, in the current study, all three stages of AKI were associated with similar rates of recovery and CKD development, suggesting that physicians should focus on the recovery of renal function, even when the initial AKI is not severe. The differing results seen in the present study, in comparison with previous reports, may reflect confounding variables, such as the severity of infection, immune function, duration of exposure to nephrotoxic drugs and timing of RRT initiation, which were not assessed in the current study.

Risk factors for CKD after AKI were seen to include older age, diabetes and higher creatinine level at discharge. Higher creatine levels at discharge indicate delayed or lack of renal function recovery, despite the resolution of the initial infection. Manish et al. suggested that early reversible AKI within the first day of admission was associated with a better survival rate than was no, new or persistent AKI [12]. By contrast, Jones et al. demonstrated that even reversible AKI is strongly associated with an increased risk of progression to CKD [24]. Identifying a direct causative mechanism between AKI and CKD may be impossible, but recent studies demonstrated that persistent kidney injury induced by septic AKI is coupled with systemic inflammation. Renal repair can lead to malfunctions in inflammation and fibrosis and vascular rarefaction that leads to continuous cell and tissue disruption [25,26]. Considering this, protein biomarkers such as neutrophil gelatinase-associated lipocalin and interleukin 6 are likely to have an important role in assessing kidney injury, and aiding the discovery of new treatment targets [26,27,28].

The current study has several limitations. First, the results are limited by the retrospective study design, and as the data were obtained from a single center, it may not be possible to generalise the findings to other populations. Secondly, because of the long-term study period, patients may not have received consistent prehospital and hospital treatment, which may have affected the outcomes. Thirdly, the diagnosis and classification of AKI based only on serum creatinine values may not have captured all relevant cases of AKI, as extremes in muscle mass or dietary protein consumption may affect serum creatinine, and may not reflect true kidney functioning [29]. A recent study demonstrated that the use of serum creatinine only, urine output only or both factors of the KDIGO criteria showed differing outcomes; for example, hospital mortality within the three groups was 9.2% (serum creatinine only), 7.5% (urine output only) and 26.7% (both serum creatinine and urine output) [30]. Moreover, previous CKD or ESRD, or patients with high risk factors (pre-existing proteinuria, albuminuria, decreased urine output, or long-term usage of medication, which could induce renal dysfunction) could be included in study population. Also, not all patients had baseline serum creatinine or eGFR, so we could not confirm to what extent septic shock was responsible for kidney dysfunction. Because our data did not have urine analysis and renal ultrasonography, which contain other important clues to diagnosis CKD development, there was a potential risk of not having an accurate total number of CKD patients upon follow up. In addition, there is a risk of CKD misclassification, as only single timepoint data were used during the follow-up. Finally, a considerable number of patients (217 of 839, 25%) were lost to follow-up, which may have impacted on the results obtained. To make up for this problem, we compared the baseline characteristics between the study group and the follow-up loss group, and there were no significant differences between the two groups. Moreover, we conducted a sensitivity analysis (Appendix A), and found that the trend of ORs for discharge creatinine did not change significantly. These results indirectly imply that follow-up loss patients did not make up a significant bias to our result. However, our results still had the possibility to change if follow-up loss data were included.

## 5. Conclusions

In conclusion, AKI severity was correlated with mortality, but it did not correlate with the development of CKD, and patients progressed to CKD, even when the initial AKI stage was not severe. Physicians should focus on the recovery of renal function, and ensure the careful follow-up of patients with risk factors for the development of CKD.

## Figures and Tables

**Figure 1 jcm-07-00554-f001:**
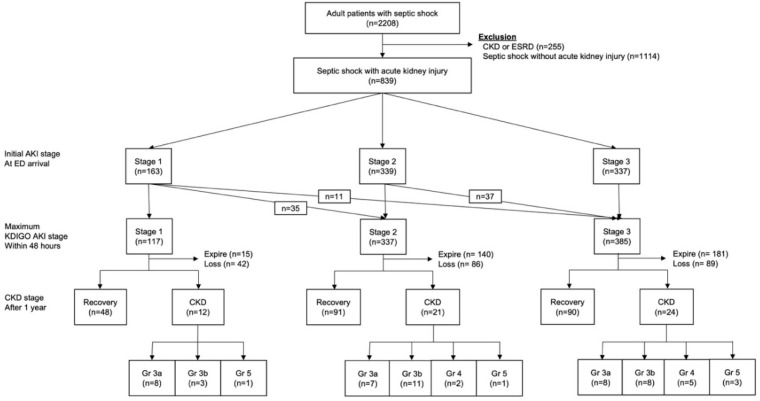
Flowchart of the study population.

**Table 1 jcm-07-00554-t001:** Characteristics of patients.

Characteristics	Total *n* = 286	Non-CKD after 1 Year *n* = 229	CKD after 1 Year *n* = 57	*p*-Value
Age	63.7 (56.0–72.0)	64.0 (55.0–70.0)	71.0 (61.3–77.8)	0.001
Male	181 (63.3)	152 (66.4)	29 (50.9)	0.033
Underlying disease				
HTN	102 (35.7)	75 (32.8)	27 (47.4)	0.045
Stroke	24 (8.4)	18 (7.9)	6 (10.5)	0.592
DM	84 (29.4)	54 (23.6)	30 (52.6)	<0.001
Coronary artery disease	22 (7.7)	14 (6.1)	8 (14.0)	0.054
Chronic pulmonary disease	29 (10.1)	22 (9.6)	7 (12.3)	0.623
Liver cirrhosis	43 (15.0)	37 (16.2)	6 (10.5)	0.312
Malignancy	79 (27.6)	56 (25.8)	20 (35.1)	0.186
Infection site				
Unknown	4 (4.1)	2 (8.8)	2 (2.7)	0.788
Pulmonary	21 (21.6)	15 (20.5)	6 (25.0)	0.776
Urinary	23 (23.7)	18 (24.7)	5 (20.8)	0.788
Gastrointestine	15 (15.5)	13 (17.8)	2 (8.3)	0.345
Hepatobiliary	30 (30.9)	21 (28.8)	9 (37.5)	0.452
Others	11 (11.3)	9 (12.3)	2 (8.3)	0.572
Laboratory				
WBC (×10^3^/uL)	10.5 (5.1–17.5)	10.5 (5.7–18.0)	10.6 (4.0–14.9)	0.491
Hb (g/dL)	11.5 (9.3–13.2)	11.9 (9.9–13.6)	14.93 (10.6–22.6)	<0.001
PLT (×10^3^/uL)	137.0 (72.5–207.0)	138.0 (75.5–207.0)	130.0 (66.25–210.5)	0.921
BUN (mg/dL)	32.0 (24.8–45.0)	31.0 (23.0–40.5)	36.0 (29.0–53.0)	0.017
Baseline Cr (mg/dL)	0.72 (0.63–0.86)	0.70 (0.61–0.82)	0.80 (0.72–0.98)	<0.001
Initial Cr (mg/dL)	1.8 (1.4–2.5)	1.8 (1.4–2.4)	2.1 (1.5–3.0)	0.037
Peak Cr (mg/dL)	2.0 (1.5–2.7)	1.9 (1.5–2.6)	2.4 (1.6–3.5)	0.002
Discharge Cr (mg/dL)	0.9 (0.7–1.1)	0.8 (0.6–1.0)	1.1 (0.9–1.8)	<0.001
Lactate (mmol/L)	3.3 (2.0–5.5)	3.3 (2.0–5.6)	3.0 (1.8–4.5)	0.389
CRP (mg/dL)	15.3 (5.9–22.2)	16.1 (6.7–22.3)	11.9 (5.13–22.0)	0.278

Data are presented as *n* (%) or median with interquartile ranges. HTN = hypertension; DM = diabetes mellitus; CKD = chronic kidney disease; WBC = white blood cells; Hb = hemoglobin; PLT = platelet; BUN = blood urea nitrogen; Cr = creatinine; CRP = c-reactive protein.

**Table 2 jcm-07-00554-t002:** Adjusted odds ratios of the AKI stage for CKD development in patients with sepsis-induced AKI.

Variables	Multivariate Analysis		
	OR	95% CI	*p*-Value
CKD Development			
Initial Cr			
KDIGO stage 1	Reference		
KDIGO stage 2	0.783	0.375–1.635	0.515
KDIGO stage 3	0.924	0.444–1.923	0.832
Maximum Cr			
KDIGO stage 1	Reference		
KDIGO stage 2	0.879	0.396–1.950	0.751
KDIGO stage 3	1.111	0.513–2.405	0.789
All-Cause Mortality			
Initial Cr			
KDIGO stage 1	Reference		
KDIGO stage 2	2.637	1.719–4.046	<0.001
KDIGO stage 3	2.933	1.913–4.499	<0.001
Maximum Cr			
KDIGO stage 1	Reference		
KDIGO stage 2	4.832	2.696–8.668	<0.001
KDIGO stage 3	5.909	3.316–10.530	<0.001

AKI = acute kidney injury; CKD = chronic kidney disease; OR = odds ratio; CI = confidence interval; Cr = creatinine; KDIGO = Kidney Disease Improving Global Outcomes.

**Table 3 jcm-07-00554-t003:** Multivariate logistic regression of factors associated with the occurrence of CKD after one year.

Variables	Univariate Analysis			Multivariate Analysis		
	OR	95% CI	*p*-Value	OR	95% CI	*p*-Value
Age	1.066	1.027–1.107	<0.001	1.070	1.033–1.108	<0.001
HTN	0.996	0.487–2.039	0.991			
DM	2.656	1.341–5.257	0.005	2.620	1.352–5.078	0.004
CAD	1.914	0.638–5.745	0.247			
LC	0.992	0.360–2.730	0.987			
Malignancy	1.250	0.601–2.600	0.551			
Hb	0.833	0.734–0.946	0.005	0.840	0.744–0.949	0.005
Discharge Cr	2.503	1.371–4.569	0.003	2.686	1.499–4.812	<0.001

Abbreviations: OR = odds ratio; CI = confidence interval; HTN = hypertension; DM = diabetes mellitus; CAD = coronary artery disease; LC = liver cirrhosis; Hb = hemoglobin; Cr = creatinine.

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
