# Peer review of "One-Year Progression and Risk Factors for the Development of Chronic Kidney Disease in Septic Shock Patients with Acute Kidney Injury: A Single-Centre Retrospective Cohort Study"

_jcm, 2018, doi:10.3390/jcm7120554_

Round 1
Reviewer 1 Report
In this study, authors investigated the risk factors in progression of CKD in septic patients. Authors suggested that physicians should monitor patient's kidney function in septic patient with AKI. I did not find any novelty in this study, these recommendation have been already in practice, therefore i would ask authors what is new in their study?
Author Response
We agreed with your opinion that several studies have reported the incidence and risk factors of the development septic AKI for early recognition, prevention, or therapeutic interventions. However, almost previous studies followed up the patients 28-90 days. It is unclear that patients who develop septic AKI completely recover renal function by hospital discharge, and these patients whether the consequences of recovery or progress are different in patients based on the stage of AKI. Understanding the effects of renal recovery at hospital discharge on long-term outcomes after AKI can help inform clinicians about prognosis and may have important implications for follow up of these patients. Moreover, most of the studies focused on the relationship between AKI and mortality. Mortality as the primary outcome was readily measurable than that of CKD development, therefore, there was a relatively small number of studies which conducted to analyze the relationship between septic AKI on CKD occurrence. We included a large number of cohort and classified based on KDIGO stages. It is well-known that mortality is proportionally increasing according to KDIGO severity, however, the relationship between AKI and CKD occurrence remains unclear. The previous study reported that AKI severity also affected the development of CKD, however, our study showed that CKD development did not proportionally increase and this result imply mild AKI could be a risk factor for CKD development and clinician would pay attention even mil AKI.
Reviewer 2 Report
Kim et al., performed a retrospective study to evaluate the long-term progression and risk factors for the development of chronic kidney disease (CKD) in 2,208 patients with septic shock with acute renal failure (AKI). The authors concluded that the severity of the AKI correlated with mortality but did not correlate with the development of CKD.
It’s an interesting study and authors have done excellent work in planning the study with good methods and analysis along with a thorough discussion.
Following are my comments and suggestions to the authors,
1.Experimental section: Page 2
- Was any attempt made to evaluate the preexisting proteinuria in the patients recruited for the study?
There is evidence that patients with AKI with proteinuria have had a higher risk of CKD than ones with out, (1)
-The authors wrote’ To evaluate CKD status after one year, serum creatinine and GFR levels were obtained after discharge from the electronic medical records of all patients, 12 ± 3 months from initial admission’.
Do any other parameters were collected to see if the patients developed CKD? Like Urine analysis for proteinuria, hematuria or casts or availability of renal USG to see changes in line with chronic kidney disease?
-The authors wrote’ When multiple records were present, the lowest serum creatinine and GFR values were recorded.’ This is interesting as the common practice is to go with the higher serum Cr and eGFR as the new baseline. Can authors explain why they have decided to go with the higher serum creatinine? Any evidence to support their use of the higher serum creatinine?
-Please disclose which formula was used to calculate the eGFR in patients? Given the timeline of the recruited patients in the study, multiple formulas might have been used to calculate eGFR? If so, what did the authors do to reduce errors? If only one formula is used, please disclose in the manuscript.
-The authors wrote’ Serum creatinine levels were assessed in all patients at least once each day during the hospital stay, and initial, peak within 48 hours and discharge values were recorded’.
There is no mention in the study of any of the studied 839 patients requiring the renal replacement therapy (RRT) such as intermittent hemodialysis or Continuous dialysis or urgent start peritoneal dialysis during their hospitalization. On those patients who required the RRT how does the discharge creatinine or eGFR is calculated?
-Under data collection, the authors wrote ‘Patients classified as G3a, G3b, G4, and G5 were included in the analysis; those with G1 and G2 disease were excluded as they were considered to be at low risk of developing CKD’. Do they mean low risk to develop ESRD?
2. Results: Page 3
- Its surprising to see only five patients ended on the ESRD after a year of follow up. Any of the patients lost to follow up because they ended up on dialysis? Authors need to clarify. In a recent trial including AKI in Septic shock patients nearly 30% of the patients required RRT, (2).
- Also, the authors need to explain why so many patients lost to follow Up which is almost 26% of the included study participants (839). The higher percentage of the patients failed to follow up can affect the results and cause significant bias.
References:
1.James MT, Hemmelgarn BR, Wiebe N, et al. Glomerular filtration rate, proteinuria, and the incidence and consequences of acute kidney injury: a cohort study. Lancet. 2010; 376:2096–2103
2.Gordon AC, Mason AJ, Thirunavukkarasu N et al. (2016) Effect of early vasopressin vs. norepinephrine on kidney failure in patients with septic shock: the VANISH randomized clinical trial. JAMA 316:509–518
Author Response
Reviewer 2 comments
1. Experimental section: Page2
- Was any attempt made to evaluate the preexisting proteinuria in the patients recruited for the study?
There is evidence that patients with AKI with proteinuria have had a higher risk of CKD than ones with out,
Response>
We agreed with the reviewer’s concern about that point. We excluded patients who had an outpatient or inpatient diagnosis of chronic kidney disease or end-stage renal disease or codes for dialysis and had a prior estimated GFR < 60 mL/min1.73m2. We reviewed our whole study cohort medical records again and found that 95 patients had no baseline urinalysis and about 250 patients had albuminuria in the baseline study group. Among survivors, 72 patients had albuminuria, and there was no statistically difference between two groups (CKD (+) vs. CKD (-); 32.7% vs. 25.7%; p = 0.384) even though there were 20 missing data. We changed sentence about exclusion criteria for the study population more detail in the materials and methods section. We also added and clarify more detail sentence in limitation part.
“We excluded individuals who were younger than 18 years and pregnant. Moreover, in addition to evaluate the effect of newly developed AKI on CKD, we also excluded patients who had diagnosis previously with CKD and end-stage renal disease (ESRD) requiring renal replacement therapy (RRT).”
“Previous CKD or ESRD who had outpatient or inpatient diagnosis of pre-existing CKD and ESRD, had code related with hemodialysis, and had a prior diagnosis of AKI or baseline estimated glomerular filtration rate (eGFR) lower than 60 mL/min/1.73m2 were identified via electronic medical records.”
- The authors wrote’ To evaluate CKD status after one year, serum creatinine and GFR levels were obtained after discharge from the electronic medical records of all patients, 12 ± 3 months from initial admission’. Do any other parameters were collected to see if the patients developed CKD? Like urine analysis for proteinuria, hematuria or casts or availability of renal USG to see changes in line with chronic kidney disease?
Response>
Thank you for your valuable suggestion. We totally agree with your opinion that the possibility of under measurement of CKD development was the main limitation of our result. Because of retrospective study design and occasional visit of the patients, we could not find all of the data that you suggest. When we designed the study, we only included eGFR based on KDIGO CKD guideline. However, besides of serum creatinine and eGFR level, we tried to found CKD development through inpatient or outpatient diagnosis of CKD or ESRD, and code related with hemodialysis via electronic medical records in order to reduce missing CKD development. We explained more detail in the materials and methods section.
“Moreover, in order to reduce missing diagnosis, we collected data of inpatient or outpatient diagnosis of CKD or ESRD, and code related with hemodialysis via medical records.”
- The authors wrote’ When multiple records were present, the lowest serum creatinine and GFR values were recorded.’ This is interesting as the common practice is to go with the higher serum creatinine and eGFR as the new baseline. Can authors explain why they have decided to go with the higher serum creatinine? Please provide any evidence to support their use of the higher serum creatinine?
Response>
We are sorry to make a reviewer confuse. We selected the highest serum creatinine and lowest eGFR values in the analysis. We changed the sentence as follow.
“When multiple records were present, the highest serum creatinine and lowest eGFR values were recorded.”
- Please disclose which formula was used to calculate the eGFR in patients? Given the timeline of the recruited patients in the study, multiple formulas might have been used to calculate eGFR? If so, what did the authors do to reduce errors? If only one formula is used, please disclose in the manuscript.
Response>
Thank you for your kind suggestion. We used all eGFR based on only one formula with Modification of Diet in Renal Disease (MDRD). GFR = 175 x Serum Cr -1.154 x age -0.203 x 1.212 (if patient is black) x 0.742 (if female). We disclosed this formula and added in the article.
“All eGFRs were calculated by MDRD (GFR = 175 x serum Cr -1.154 x age -0.203 x 1.212 (if patient is black) x 0.742 (if female)).”
- The authors wrote’ Serum creatinine levels were assessed in all patients at least once each day during the hospital stay, and initial, peak within 48 hours and discharge values were recorded’. There is no mention in the study of any of the studied 839 patients requiring the renal replacement therapy (RRT) such as intermittent hemodialysis or continuous dialysis or urgent start peritoneal dialysis during their hospitalization. On those patients who required the RRT how does the discharge creatinine or eGFR is calculated?
Response>
Thank you for your suggestion. Among 839 patients, 151 patients had continuous RRT during admission and 6 patients needed intermittent hemodialysis after stopping continuous RRT. We added sentences in baseline characteristics in the result section. On the other hand, we calculated eGFR of patients requiring RRT with MDRD as those who did not require RRT. RRT may influence the level of creatinine between CKD and non-CKD group. However, the efficacy of RRT is out of scope in this study, and the impact of the influence of RRT on eGFR or creatinine level was not significant because the number of patients who needed RRT was relatively small (n = 6).
“Among them, 151 patients applied continuous RRT during admission and 6 patients needed intermittent hemodialysis after stop continuous RRT.”
- Under data collection, the authors wrote ‘Patients classified as G3a, G3b, G4, and G5 were included in the analysis; those with G1 and G2 disease were excluded as they were considered to be at low risk of developing CKD’. Do they mean low risk to develop ESRD?
Response>
Thank you for your kind correction. It was a typo error. We’ve changed the word CKD to ESRD.
“Patients classified as G3a, G3b, G4, and G5 were included in the analysis; those with G1 and G2 disease were excluded as they were considered to be at low risk of developing ESRD.”
2. Results: Page 3
- Its surprising to see only five patients ended on the ESRD after a year of follow up. Any of the patients lost to follow up because they ended up on dialysis? Authors need to clarify. In a recent trial including AKI in Septic shock patients nearly 30% of the patients required RRT, (References: Gordon AC, Mason AJ, Thirunavukkarasu N et al. (2016) Effect of early vasopressin vs. norepinephrine on kidney failure in patients with septic shock: the VANISH randomized clinical trial. JAMA 316:509–518).
Response>
In VANISH study, the incidence of AKI required RRT was about 30%. In our study also the incidence of AKI stage 3 was 40.17%. Unfortunately, VANISH study observed only a 28day follow up unlike ours of 1 year. Moreover, our study excluded those who had previously diagnosed ESRD patients or CKD patients which are a well-known risk factor for AKI. One possible reason for low incidence of ESRD after a year of follow up was that among died patients within one year, ESRD would have occurred. However, because of our retrospective cohort design, we could not get whole data.
- Also, the authors need to explain why so many patients lost to follow up which is almost 26% of the included study participants (839). The higher percentage of the patients failed to follow up can affect the results and cause significant bias
Response>
We totally agree reviewer’s opinion that our study has a high proportion of the patients failed to follow up and this can affect the results. Among 217 follow up loss patients, KDIGO AKI stage 1 patients were 42 (36%), stage 2 was 86 (26%), and stage 3 was 89 (23%). To make up this problem, we compared baseline characteristics between the study group and follow up loss group and there were no significantly different between two groups. Moreover, we conducted sensitivity analysis (see the table below), and found that the trend of ORs for discharge creatinine did not change significantly. These results indirectly imply that follow-up loss patients did not make significant bias in our result. We added this sentence in discussion and table in supplement data.
“To make up this problem, we compared baseline characteristics between the study group and follow up loss group and there were no significantly different between two groups. Moreover, we conducted sensitivity analysis (see the table below), and found that the trend of ORs for discharge creatinine did not change significantly. These results indirectly imply that follow-up loss patients did not make significant bias to our result.”

Reviewer 3 Report
1. There was no detailed description why you define the maximum KDIGO stage would occur within 48 hours.
2. There were so many patients were lost to followed-up in your study and the amount was large enough to influence the outcome of this study.
3. Mortality and CKD progression are competing event of outcome, patients those who expired during this period would not progress to CKD after 1 year. I suggest that you can perform competing risk regression model to declare it.
4. There was no patient with CKD in table 1. However, the initial Cr showed 1.8 1.4-2.5) mg/dL and which is an abnormal Cr level in most lab data as we known. Was the initial Cr level was the baseline Cr?
5. The mortality you mentioned in the discussion was 26%. I can’t see how to get the result.
6. The precise definition of mortality in this study is unclear.
7. The patient number was so different between CKD and non-CKD group.
Thank you.
Author Response
Reviewer 3 comments
1. There was no detailed description why you define the maximum KDIGO stage would occur within 48 hours.
Response>
According to KDIGO AKI guideline, AKI is defined as increase in serum creatinine within 48 hours. We tried to follow this guideline and wanted to check the extent of progression. We postulated that the progression was more meaningful than that of initial value.
2. There were so many patients were lost to followed-up in your study and the amount was large enough to influence the outcome of this study.
Response>
As we mentioned in the previous comments (reviewer 2), we totally agree reviewer’s opinion that our study has a high proportion of the patients failed to follow up and this can affect the results. Among 217 follow up loss patients, KDIGO AKI stage 1 patients were 42 (36%), stage 2 was 86 (26%), and stage 3 was 89 (23%). To make up this problem, we compared baseline characteristics between the study group and follow up loss group and there were no significantly different between two groups. Moreover, we conducted sensitivity analysis (see the table below), and found that the trend of ORs for discharge creatinine did not change significantly. These results indirectly imply that follow-up loss patients did not make significant bias in our result. We added this sentence in discussion and table in supplement data.
“To make up this problem, we compared baseline characteristics between the study group and follow up loss group and there were no significantly different between two groups. Moreover, we conducted sensitivity analysis (see the table below), and found that the trend of ORs for discharge creatinine did not change significantly. These results indirectly imply that follow-up loss patients did not make significant bias to our result.”
Supplement1. Univariate logistic regression of factors associated with the occurrence of CKD after 1 year including follow-up loss patients (sensitivity analysis)
Discharge Creatine level | Univariate analysis | ||||
OR | 95% CI | p | |||
All CKD | 1.915 | 1.351 – 2.714 | < 0.001 | ||
No CKD | 1.550 | 1.174 – 2.047 | 0.002 | ||
Partial CKD (20% incidence) | 1.449 | 1.120 – 1.874 | 0.005 | ||
Abbreviations: OR = odds ratio; CI = confidence interval; CKD = chronic kidney injury.
3. Mortality and CKD progression are competing event of outcome, patients those who expired during this period would not progress to CKD after 1 year. Please you perform competing risk regression model to declare it.
Response>
We totally agree to the reviewer’s opinion. Death may be the most important adverse events of septic shock survivor and septic AKI is one of most life-threatening manifestations of septic shock. In this study, we tried to reveal the relationships between AKI and its renal complication itself, CKD, without considering death. According to your suggestion, we tried to perform competing risk regression model to find how much death hinder CKD occurrence. However, contrary to time of death, it is impossible to know the certain time of CKD occurrence because of irregular time interval when each patient visits outpatients or inpatients department.
Instead of performing competing risks methods, we analyzed our data with changing primary outcome to the composite outcome including death and CKD and compared the results to make an inference the effect of death (Table 1). The results show that the trends and main finding of our study does not change although the extent of some ORs change. This imply that including death patients or not significant impact our data.
Table 1. Characteristics of Patients for all adverse events (death and CKD)
Characteristics | Total n = 839 | No adverse events n = 440 | Adverse events n = 399 | p-value |
Age | 67.0 (58.0 – 74.0) | 65.0 (57.0 – 73.0) | 67.0 (60.0 – 75.0) | 0.013 |
Male | 529 (63.1) | 272 (61.8) | 257 (64.4) | 0.474 |
Underlying disease | ||||
HTN | 304 (36.2) | 162 (36.8) | 142 (35.6) | 0.720 |
Stroke | 60 (7.2) | 34 (7.7) | 26 (6.5) | 0.506 |
DM | 236 (28.2) | 111 (25.2) | 125 (31.4) | 0.054 |
CAD | 73 (8.7) | 29 (6.6) | 44 (11.0) | 0.027 |
CPD | 93 (11.1) | 37 (8.4) | 56 (14.0) | 0.011 |
Liver cirrhosis | 127 (15.1) | 59 (13.4) | 68 (17.0) | 0.149 |
Malignancy | 326 (38.9) | 157 (35.7) | 169 (42.4) | 0.055 |
Infection site | ||||
Unknown | 47 (5.6) | 22 (5.0) | 25 (6.3) | 0.455 |
Pulmonary | 167 (19.9) | 67 (15.2) | 100 (25.1) | <0.001< span=""> |
Urinary | 103 (12.3) | 65 (14.8) | 38 (9.5) | 0.027 |
Gastrointestine | 84 (10.0) | 48 (10.9) | 36 (9.0) | 0.420 |
Hepatobiliary | 157 (18.7) | 91 (20.7) | 66 (16.5) | 0.132 |
Others | 8 (1.0) | 3 (0.7) | 5 (1.3) | 0.488 |
Laboratory | ||||
WBC (×103/uL) | 10.6 (4.8 – 18.3) | 11.3 (6.3 – 18.5) | 10.1 (3.4 – 17.1) | 0.022 |
Hb (g/dl) | 10.8 (9.0 – 12.8) | 11.1 (9.3 – 13.2) | 10.4 (8.6 – 12.2) | <0.001< span=""> |
PLT (×103/uL) | 137.0 (68.0 – 212.0) | 14.0 (85.5 – 226.0) | 128.0 (57.0 – 202.0) | 0.004 |
BUN (mg/dL) | 34.0 (25.0 – 49.0) | 32.0 (24.0 – 44.0) | 38.0 (26.0 – 53.0) | <0.001< span=""> |
Initial Cr (mg/dL) | 1.89 (1.45 – 2.70) | 1.81 (1.42 – 2.52) | 2.02 (1.52 – 3.00) | 0.001 |
Peak Cr (mg/dL) | 2.04 (1.54 – 2.86) | 1.92 (1.50 – 2.70) | 2.19 (1.70 – 3.06) | <0.001< span=""> |
Discharge Cr (mg/dL) | 0.97 (0.70 – 1.56) | 0.80 (0.62 – 1.03) | 1.34 (0.90 – 2.17) | <0.001< span=""> |
Initial lactate (mmol/L) | 3.6 (2.1 – 6.0) | 3.4 (2.0 – 5.7) | 3.9 (2.2 – 6.5) | 0.009 |
CRP (mg/dL) | 15.9 (6.6 – 22.8) | 15.8 (6.9 – 23.0) | 15.9 (6.3 – 22.6) | 0.755 |
Data are presented as n (%) or median with interquartile ranges.
Abbreviations: HTN = hypertension; DM = diabetes mellitus; CAD = coronary artery disease; CPD = chronic pulmonary disease; WBC = white blood cells; Hb = hemoglobin; PLT = platelet; BUN = blood urea nitrogen; Cr = creatinine; CRP = c-reactive protein.
Table 7. Adjusted odds ratios of AKI stage for all adverse events (death, CKD) in patients with sepsis-induced AKI
Variables | Multivariate analysis | ||
OR | 95% CI | p-value | |
Initial | |||
KDIGO stage 1 | Reference | ||
KDIGO stage 2 | 2.015 | 1.363 – 2.979 | < 0.001 |
KDIGO stage 3 | 2.351 | 1.590 – 3.478 | < 0.001 |
Maximum | |||
KDIGO stage 1 | Reference | ||
KDIGO stage 2 | 3.123 | 1.932 – 5.047 | < 0.001 |
KDIGO stage 3 | 3.958 | 2.463 – 6.362 | < 0.001 |
Abbreviations: AKI = acute kidney injury; CKD = chronic kidney disease; OR = odds ratio; CI = confidence interval; KDIGO = Kidney Disease Improving Global Outcomes.
Table 8. Multivariate logistic regression of factors associated with occurrence of CKD after 1 year
Variables | Univariate analysis | Multivariate analysis | ||||
OR | 95% CI | p | OR | 95% CI | p | |
Male | 0.846 | 0.611 – 1.172 | 0.315 | |||
CPD | 1.580 | 0.943 – 2.648 | 0.085 | |||
LC | 1.090 | 0.694 – 1.713 | 0.708 | |||
Malignancy | 1.539 | 1.117 – 2.120 | 0.008 | 1.482 | 1.081 – 2.032 | 0.015 |
Pneumonia | 1.606 | 1.067 – 2.417 | 0.023 | 1.763 | 1.205 – 2.579 | 0.003 |
UTI | 0.788 | 0.488 – 1.273 | 0.331 | |||
WBC | 0.998 | 0.984 – 1.012 | 0.754 | |||
Hb | 0.931 | 0.881 – 0.985 | 0.013 | 0.923 | 0.874 – 0.975 | 0.004 |
PLT | 0.999 | 0.998 – 1.001 | 0.228 | |||
Discharge Cr | 2.594 | 2.074 – 3.245 | <0.001< span=""> | 2.599 | 2.081 – 3.245 | <0.001< span=""> |
Lactate | 1.057 | 1.007 – 1.108 | 0.024 | 1.055 | 1.006 – 1.106 | 0.027 |
Abbreviations: OR = odds ratio; CI = confidence interval; CPD = chronic pulmonary disease; LC = liver cirrhosis; UTI = urinary tract infection; WBC = white blood cell; Hb = hemoglobin; PLT = platelet; Cr = creatinine.
4. There was no patient with CKD in table 1. However, the initial Cr showed 1.8 1.4-2.5) mg/dL and which is an abnormal Cr level in most lab data as we known. Was the initial Cr level was the baseline Cr?
Response>
Thank you for your good suggestion. Initial Cr level was measured on admission when AKI was diagnosed. To clarify the meaning as the reviewer’s opinion, we provided baseline Cr of the study population in Table 1 and added the sentence as follow. Even though baseline Cr was higher in CKD (+) group, it was within normal range (0.8 (inter-quartile range 0.72 – 0.98)).
“Regarding laboratory values, baseline, initial, peak and discharge creatinine, blood urea nitrogen and hemoglobin levels tended to be higher in the CKD group than in the non-CKD group.”
5. The mortality you mentioned in the discussion was 26%. I can’t see how to get the result.
Response>
Thank you for exact correction. It was typo and the number 40% (336/839) was correct. We change the sentence in discussion section.
“Secondary outcomes included all-cause mortality and RRT dependence within the 1-year follow-up period.”
“Long-term all-cause mortality was 40% (336/839), and while the severity of KDIGO AKI stage correlated with mortality, it did not correlate with the development of CKD.”
6. The precise definition of mortality in this study is unclear.
Response>
To clarify the meaning, we changed the word from mortality to all-cause mortality. The date of patient’s death was obtained from the National Health Insurance Service in South Korea. We added this sentence in material and method section.
“The date of patient’s death was extracted from the National Health Insurance Service in South Korea.”
7. The patient number was so different between CKD and non-CKD group
Response>
Thank you for your good suggestion. We performed retrospective cohort study and number of CKD and non-CKD group were observed during our study period. Even though there were differences in number, these number of patients were enough to perform statistics.

Round 2
Reviewer 1 Report
Authors have satisfactorily answered my question, I have no further comment.
Author Response
Thank you for your generous review.
Reviewer 2 Report
I have reviewed the original manuscript and am overall satisfied with authors responses and modifications to my comments and suggestions.
I have one final comment/suggestion for the authors.
-Under the discussion, please mention the limitation of not having access to the Urine analysis to check for other signs of CKD in terms of unexplained proteinuria, hematuria, casts or the availability of renal USG to see changes in line with chronic kidney disease(CKD) and potential risk of not having accurate total number of CKD patients on follow up.
I thank the authors for submitting the manuscript to JCM and wish them all luck.
Author Response
-Under the discussion, please mention the limitation of not having access to the Urine analysis to check for other signs of CKD in terms of unexplained proteinuria, hematuria, casts or the availability of renal USG to see changes in line with chronic kidney disease(CKD) and potential risk of not having accurate total number of CKD patients on follow up.
Response>
According to the reviewer’s suggestion, we add this in the limitation section as below
“Because our data had not all urine analysis and renal ultrasonography, which had other important clues to diagnosis CKD development, there was a potential risk of not having an accurate total number of CKD patients on follow up.”
Reviewer 3 Report
In your conclusion, you mentioned that the severity of AKI was not correlated with development of CKD but correlated with mortality. However, I did not see the statistical result support this conclusion. No comparison of the relation between AKI severity and mortality was noted.
What't the definition and different of "initial" Cr and "baseline" Cr?
You try to make up the problem of considerable number of patients loss of follow-up by comparing the "loss" group and study group. However, even though the baseline characteristics were similar, the result could be different after analysis for the proportion of missing data was too much.
Line 84: newly developed AKI on CKD, did you mean developed AKI "to" CKD?
Line 234: Type error: fdiabetes
Author Response
In your conclusion, you mentioned that the severity of AKI was not correlated with development of CKD but correlated with mortality. However, I did not see the statistical result support this conclusion. No comparison of the relation between AKI severity and mortality was noted.
Response>
Thank you for your great suggestion. We’ve added this on table 2 as follow.
“The adjustedORs of the initial and maximum KDIGO AKI stage for CKD development and all-cause mortality within 1 year are shown in Table 2.”
“Meanwhile, all-cause mortality was proportionally increased according to KDIGO classification by initial and maximum Cr level.”
Table 2. Adjusted odds ratios of AKI stage for CKD development and all-cause mortality in patients with sepsis-induced AKI
Variables | Multivariate analysis | ||
OR | 95% CI | p-value | |
CKD development | |||
Initial Cr | |||
KDIGO stage 1 | Reference | ||
KDIGO stage 2 | 0.783 | 0.375 – 1.635 | 0.515 |
KDIGO stage 3 | 0.924 | 0.444 – 1.923 | 0.832 |
Maximum Cr | |||
KDIGO stage 1 | Reference | ||
KDIGO stage 2 | 0.879 | 0.396 – 1.950 | 0.751 |
KDIGO stage 3 | 1.111 | 0.513 – 2.405 | 0.789 |
All-cause mortality | |||
Initial Cr | |||
KDIGO stage 1 | Reference | ||
KDIGO stage 2 | 2.637 | 1.719 – 4.046 | < 0.001 |
KDIGO stage 3 | 2.933 | 1.913 – 4.499 | < 0.001 |
Maximum Cr | |||
KDIGO stage 1 | Reference | ||
KDIGO stage 2 | 4.832 | 2.696 – 8.668 | < 0.001 |
KDIGO stage 3 | 5.909 | 3.316 – 10.530 | < 0.001 |
AKI = acute kidney injury; CKD = chronic kidney disease; OR = odds ratio; CI = confidence interval; Cr = creatinine; KDIGO = Kidney Disease Improving Global Outcomes.
What't the definition and different of "initial" Cr and "baseline" Cr?
Response>
We are sorry to make confusion to the reviewer. Initial Cr means the Cr level measured at admission when patients were suffered from septic shock with AKI. Baseline Cr means the Cr level measured preceding 1 week to 1 year before admission. Among 839 patients, most of our cohort (n = 780) had baseline Cr and only 58 patients had no baseline Cr. And all of these patients had no underlying diseases related to kidney disease and we postulated that baseline was normal for that patient. We added sentences in method and discussion.
“Serum creatinine levels were assessed in all patients at least once each day during the hospital stay, and baseline, initial, peak within 48 hours and discharge values were recorded. Baseline creatinine level was measured preceding 1 week to 1 year before admission, and the initial level was measured at admission.”
“And not all patients had baseline serum creatinine or eGFR, we could not confirm to what extent septic shock was responsible for the kidney dysfunction.”
You try to make up the problem of considerable number of patients loss of follow-up by comparing the "loss" group and study group. However, even though the baseline characteristics were similar, the result could be different after analysis for the proportion of missing data was too much.
Response>
We totally agree with your concern. Considerable number of follow-up loss patient was the major limitation of the retrospective cohort study. However, our sensitivity analysis (supplement 1) showed that the trends of our results (ORs) were not changed and these indirectly proven follow-up loss data did not critical effect to our result. Never the less, we agreed your opinion and further discussed on our manuscript and stressed this in limitation section.
“However, our results had still possibility to change if follow-up loss data included.”
Supplement1. Univariate logistic regression of factors associated with the occurrence of CKD after 1 year including follow-up loss patients (sensitivity analysis)
Discharge Creatine level | Univariate analysis | ||||
OR | 95% CI | p | |||
All CKD | 1.915 | 1.351 – 2.714 | < 0.001 | ||
No CKD | 1.550 | 1.174 – 2.047 | 0.002 | ||
Partial CKD (20% incidence) | 1.449 | 1.120 – 1.874 | 0.005 | ||
Abbreviations: OR = odds ratio; CI = confidence interval; CKD = chronic kidney injury.
Line 84: newly developed AKI on CKD, did you mean developed AKI "to" CKD?
Response>
Sorry for your inconvenience. We correct typo-error.
“We excluded individuals who were younger than 18 years and pregnant. Moreover, in addition to evaluate the effect of newly developed AKI to CKD, we also excluded patients who had previously diagnosis with CKD and end-stage renal disease (ESRD) requiring renal replacement therapy (RRT).”
Line 234: Type error: fdiabetes
Response>
Thank you for your kind correction. We correct typo-error.
“Riskfactors for CKD after AKI were seen to include older age, female sex, diabetes and higher creatinine level at discharge.”
